# Thermoradiationally Modified Polytetrafluoroethylene as a Basis for Membrane Fabrication: Resistance to Hydrogen Penetration, the Effect of Ion Treatment on the Chemical Structure and Surface Morphology, Evaluation of the Track Radius

**DOI:** 10.3390/membranes13010101

**Published:** 2023-01-12

**Authors:** Lev Vladimirovich Moskvitin, Ol’ga Alekseevna Koshkina, Sergei Vital’evich Slesarenko, Mikhail Aleksandrovich Arsentyev, Leonid Izrailevich Trakhtenberg, Sergei Mikhailovich Ryndya, Eldar Parpachevich Magomedbekov, Alexander Sergeevich Smolyanskii

**Affiliations:** 1High Energy Chemistry and Radioecology Department, D. Mendeleev University of Chemical Technology of Russia, Moscow 125047, Russia; 2Technology Department, Quantum R LLC, Moscow 125319, Russia; 3Laboratory of Functional Nanocomposites, N.N. Semenov Federal Research Center for Chemical Physics, Russian Academy of Science, Moscow 119991, Russia; 4Laboratory of Chemical Kinetics, Chemical Department, Lomonosov Moscow State University, Moscow 119991, Russia; 5Department of Chemical Physics, Moscow Institute of Physics and Technology (State University), Dolgoprudny 141700, Russia; 6Laboratory of Integrated Technology of Semiconductor Devices of the Center for Radio Photonics and Microwave Technologies of the Institute of Nanotechnologies in Electronics, Spintronics and Photonics of National Research Nuclear University MEPhI, Moscow 115409, Russia

**Keywords:** modification, polytetrafluoroethylene, accelerated ions, cross-linking, modelling, fabrication of the pore structure

## Abstract

A study of the properties of thermoradiationally modified polytetrafluoroethylene and its importance for use as the basis of polymer membranes is presented. The hydrogen permeability of a TRM-PTFE film was studied in comparison with an original PTFE film, and showed a three-fold decrease in hydrogen permeability. Further, TRM-PTFE films were irradiated with accelerated Xe ions with an energy of 1 MeV with fluences from 1 × 10^8^ to 1 × 10^11^. The changes induced by ion treatment were analyzed by infrared spectroscopy of disturbed total internal reflection (IR-ATR) and by atomic force microscopy (ASM). IR-ATR indicated the absence of destruction in the fluence range from 1 × 10^8^ to 3 × 10^10^ cm^−2^ (in the area of isolated tracks) and the beginning of overlap of latent tracks on fluences from 3 × 10^10^ to 1 × 10^11^ cm^−2^. Topographic images with AFM showed layered lamellar structures that collapsed at a fluence of 10^8^ cm^−2^. The destruction was accompanied by a decrease in roughness about seven times the size of the track core observed by the ASM method, fully corresponding to the value obtained on the basis of calculations using modeling in an SRIM program.

## 1. Introduction

Fluoropolymers are considered promising materials for applications in various industries and chemical technology due to their high-performance characteristics, such as chemical resistance, electrical stability, and excellent mechanical properties manifested over a wide temperature range and at different humidity levels. One promising way of using them is by the manufacture of thin-film nanofiltration membranes [1,2]. Among the most widespread types of fluoropolymers is polytetrafluoroethylene (PTFE). Among its most significant disadvantages are cold flow resistance, low radiation resistance, and microporosity [3].

High-temperature radiation modification of PTFE in the melt should is as a promising method to eliminate the above drawbacks [4,5,6]. Therefore, to justify the possibility of using thermoradiationally modified PTFE (TRM-PTFE) as a membrane material, it is important to study the permeability of various gases and moisture through the modified polymer.

It should be noted that radiation exposure can change the gas transmission characteristics of PTFE and TRM-PTFE [7]. In the case of irradiation of amorphous-crystalline polymers, it was found that the processes of destruction of polymer chains can lead to both amorphization and an increase in the degree of crystallinity of the polymer [8,9]. Accordingly, in the first case, an increase in the gas permeability of the irradiated polymer should be expected, and in the second, a decrease in its gas transmission characteristics [10,11,12]. In addition, possible reactions between hydrogen molecules propagating in the volume of the polymeric membrane and radicals, double bonds, and other products of radiolysis, which are formed under the action of ionising radiation, cannot be ignored [13,14,15]. Consequently, there is a need to establish a correlation between the primary radiation-chemical processes and the gas transmission properties of PTFE and TRM-PTFE.

Irregularities of radiation-chemical processes occurring at high-temperature radiation modification of PTFE have been investigated mainly for ionising radiation with low linear energy transfer (LET ≈ 0.2 eV/nm; e.g., soft and hard X-ray radiation), photons of γ-radiation of isotope ^60^Co and electron fluxes with energies from 0.5 to 3 MeV [16,17,18]. At the same time, the impact of accelerated heavy ion fluxes on PTFE and TRM-PTFE is characterised by high energy emission (LFE ≈ 1000–5000 eV/nm [19]) that results in temperature increase in the latent track volume (LT) up to 104 °C [20]. During ion irradiation in the area of LT and adjacent polymer layers, high-temperature radiation modification of PTFE also takes place. Therefore, it is of interest to investigate the properties and structural changes of ion-irradiated TRM-PTFE.

The aim of this study was to investigate hydrogen permeability and patterns of radiation-induced changes in surface structure and chemistry of initial and accelerated xenon flux irradiated PTFE films. This was used to substantiate the possibility of applications of this material as a base for proton conducting membranes.

## 2. Materials and Methods

### 2.1. Method of Manufacturing Thermoradiationally Modified Polytetrafluoroethylene

Samples of block PTFE grade F-4 (GOST 1007-80) produced by GaloPolymer Kirovo-Chepetsk Ltd. (Kirovo-Chepetsk, Kirov region, Russian Federation) in the form of 100 × 100 × 10 mm plates were used for manufacturing TRM-PTFE. The samples were placed in a radiation chemical apparatus (RCA) and exposed to gamma rays of the isotope ^60^Co at the scientific installation “Gammatok” [21]. Before the start of the radiation treatment, dosimetry was carried out using the Fricke method [22]. The irradiation temperature varied from 323 to 350 °C, nitrogen was used as the external medium, irradiation was carried out at a dose of 50 kGy at a dose rate of ~3 Gy/s. Corrections for the difference between the electron densities of the Fricke dosimeter and PTFE were taken into account during the radiation treatment (the absorbed dose conversion factor for PTFE was 0.87 [23]).

It should be noted that the radiation treatment of PTFE was carried out without placing an absorber layer around the samples, which is necessary to fulfil the “electronic equilibrium” condition [24]. Without this condition the absorbed dose distribution over the PTFE sample volume is inhomogeneous.

In this case, in the volume of the PTFE plate is first be an increase in energy absorption (absorbed dose) with increasing distance from the surface to the central part of the PTFE sample being irradiated. After a certain material thickness t, which is called the thickness of the charged particle equilibrium, the radiation energy absorption decreases [24,25]. The PTFE plate thickness required to achieve maximum energy absorption is t = 2.5 mm [23]. The electron density can be calculated using the formula:(1)$$n=\left(\frac{\rho N_{A}}{M}\right)\sum_{i}Z_{i}$$
where *n* is the electron density, m^−3^; *p* is the density of PTFE (2.15 kg/m^3^ for the initial PTFE [2,3]); *N_A_* = 6.023 × 10^23^ mol^−1^ is the Avogadro constant, *M* is the molar mass, kg/mol, *Z_i_* is the atomic number, and Σ*_i_Z_i_* is the total number electrons per molecule. For PTFE, the electron density n is 6.47 × 10^23^ m^−3^, and for water is 3.23 × 10^23^ m^−3^. According to GOST 27602-88, when using water as an absorber at an ionizing radiation energy of ~1.1 MeV, the value t _equ._≈ 0.5 cm. Considering that the value of the ratio of n_PTFE_/n_H2O_ is ≈ 2.0, in the case of PTFE, the value of t_equ._≈ 0.25 cm.

Consequently, γ-irradiation of PTFE samples with dimensions less than 0.5 cm was done under conditions corresponding to the condition of electronic equilibrium. Irradiation of PTFE billets with dimensions exceeding 0.5 cm would lead to non-uniform distribution of the absorbed dose in the polymer volume. Thus, the thickness of PTFE sample at which inhomogeneity of absorbed dose distribution would be 10% and 25% is equal to t_10%_~0.75 cm and t_25%_~1.75 cm, respectively (GOST 27602-88). Since the error value of dose rate measurement with the Fricke dosimeter is ~20% [22], it can be assumed that the field of dose rate when irradiating PTFE samples with thicknesses up to 1 cm is homogeneous, and the rule of “electronic equilibrium” is satisfied in approximation.

After cooling to room temperature, the TRM-PTFE plates were removed from the RCA and the radiation-modified polymer was tested for compliance with the requirements of TU 2213-004-19978458-2021. The samples were then cut into a 40 µm thick films and reduced in thickness to 20 µm by orientation stretching on hot rollers.

### 2.2. Processing Accelerated Xenon Ion Flow

Ion modification of samples of TPM-PTFE films was carried out on a cyclic heavy ion implanter IC-100 at the Flerov Laboratory of Nuclear Physics of the Joint Institute for Nuclear Research (Dubna, Moscow Oblast, Russia) [26]. The process was carried out at a vacuum of 10^−8^ Torus and room temperature. The treatment was carried out with accelerated ^132^Xe^23+^ ions with an energy of 1 MeV/nucleon in the fluence range from 10^8^ to 10^11^. After ion irradiation was completed, TRM-PTFE samples were stored for one month in the dark, in the air, at room temperature.

### 2.3. Measurement Methods

Hydrogen permeability measurements of TRM-PTFE film samples were carried out three times in an EFC-25-01 (ElectroChem Inc., Woburn, MA, USA) sealed cell with 25 cm^3^ separated gas spaces. Viton gaskets were used as seals. On one side of the film, hydrogen was supplied at a rate of 80 mL/min, and air at a rate of 100 mL/min was supplied on the other side. The gas flows were set by Bronkhorst EL-flow gas flow controllers (Ruurlo, The Netherlands). Concentration of hydrogen in the air flow passing through the cell was measured by GG-H2-EC-10000 CTI potentiometric sensor (Calibration Technologies Inc., Columbia, SC, USA), which was calibrated in the hydrogen concentration range of 100–10,000 ppm at the air-hydrogen flow rate of 100 mL/min just before the measurement. Permeability was calculated according to the formula:(2)$$\Phi =\frac{Fl}{S\Delta p}$$
where *l* is the thickness of the film, *S* is the area of the film, *F* is the flow of hydrogen, and *∆p* is the difference in hydrogen pressure. The measurement results were compared with data on hydrogen permeability obtained during the study of samples of the initial PTFE film with a thickness of 50 microns under similar conditions (GOST 24222-80).

The infrared spectra of the disturbed total internal reflection were measured on a Shimadzu IR Prestige-21/FTIR-8000 (Shimadzu, Kyoto, Japan) spectrophotometer equipped with a horizontal attenuated total reflection ATR-8200H/8200HA attachment. The angle of incidence of the IR probe beam was 45°, and the prism material was ZnSe.

The surface morphology of the initial and ion-irradiated TRM-PTFE films was measured by atomic force microscopy using a semi-contact method with a resolution of 512 and a frequency of 1 Hz using an INTEGRA-Terma scanning probe microscope (NT-MDT Spectrum Instruments, Moscow, Russia).

### 2.4. Processing of Measurement Results

Scilab software, version 6.1 (Dassault Systemes, Velizy-Villacoublay, France) was used to process the registered IR-ATR [27].

The processing of images of the surface of the initial and xenon ion irradiated TRM-PTFE films obtained by the AFM method was carried out using both the proprietary software of the INTEGRA-Terma scanning probe microscope and Gwyddion software, which is publicly available on the Internet.

Simulation of the interaction of accelerated ions with the material was carried out using SRIM 2013 PRO software [28], which is in the public domain.

## 3. Results

### 3.1. Comparative Study of the Permeability of Films of the Initial Polytetrafluoroethylene and Thermoradiationally Modified Polymer by Hydrogen

Measurements of the hydrogen permeability of TRM-PTFE and films of the initial PTFE showed that the combined effect of gamma radiation and high temperatures led to a significant decrease in the gas permeability of PTFE (Table 1). The established values of hydrogen permeability for PTFE films were in good agreement with the previously obtained results (3.3 × 10^9^ mol·m^−1^·s^−1^·MPa^−1^ [29]).

As was noted [30,31], the formation of numerous cross-links, changes in the degree of crystallinity, and the appearance of oxygen-containing groups and peroxide macroradicals on the surface and in the polymer volume lead to a decrease in the value of gas diffusion coefficient in irradiated polymers, including TRM-PTFE. The irregularity of absorbed dose field arising during thermoradiationally modification of PTFE, and the possibility of localization of formed cross-links in the interface between the crystalline and amorphous phase, may lead to the formation of regions with increased density of cross-linked PTFE macromolecules—the so called “physical” and “chemical” cross-linking nodes [32]. This leads to a decrease in the hydrogen diffusion coefficient due to a decrease in the number of possible trajectories in the polymer volume, through which diffusion takes place, and an increase in the average path length of the diffusing molecules. To quantitatively describe the obtained results within the framework of the fractal model of gas transfer processes in irradiated polymers [33] additional studies are needed.

In addition, the detected effect of a significant deterioration of the hydrogen permeability of TRM-PTFE films may be associated with the elimination of the micropore system as a result of the combined action of γ-irradiation and high temperatures [23].

### 3.2. Investigation by Infrared Spectroscopy of the Disturbed Total Internal Reflection of the Chemical Composition of the Surface of the Initial and Accelerated Xenon Ion Irradiated Films of Thermoradiationally Modified Polytetrafluoroethylene

In the IR-ATR spectra of pristine and ion-irradiated TRM-PTFE films, the C-F bond valent vibrations located in the spectral range of 1050–1400 cm^−1^ exhibit maximum intensity (Figure 1). The maxima of these bands are located at ~1140 and ~1200 cm^−1^, which corresponds to the antisymmetric and symmetric valence vibrations of the C-F bond (Refs. [33,34] Table 2).

It should be noted that the spectral contour of CF vibrational modes recorded in the present study differed from the spectral shape of similar lines found in (Refs. [35,36,37,38] Figure 1). The reason for the discrepancy is that CF bonds are part of different atomic groups present in PTFE macromolecules (-CF_2_, -CF_3_, -CFO, -CF=CF-, etc.), and the concentration of the said atomic groups depends on the manufacturing and processing conditions of TRM-PTFE.

As a result of approximation of spectral shapes of ~1140 and ~1200 cm^−1^ bands by a set of random Voigt functions for TRM-PTFE not subjected to ion irradiation, it was found that they consist of three components with maxima at 1147 ± 0.10 (I), 1206.61 ± 0.67 (II) and 1237.93 ± 0.88 cm^−1^ (III) (Figure 2a). The nature of the vibrational mode III is related to the valence vibrations of the C-C bonds (Table 2) [39].

Exposure to a flux of xenon ions in the region of 10^8^ to 10^11^ cm^−2^ led to a broadening of the observed vibrational modes (Figure 1 and Figure 2b,c). This was due to the appearance of a new vibrational mode with a maximum at 1114–1116 cm^−1^ which was clearly observed when the IR-ATR spectrum was approximated with a set of Voigt curves in the region of 900–1400 cm^−1^ for samples irradiated with Xe ions to fluences of 10^8^, 3 × 10^10^ and 10^11^ cm^−2^. However, a vibrational mode of 1114–1116 cm^−1^ was not detected by approximating the IR-ATR spectrum for TRM-PTFE films irradiated to a fluence of 10^9^ cm^−2^.

Upon reaching a fluence of 10^11^ cm^−2^, a shift of the vibrational mode maximum of 1114–1116 cm^−1^ to 1123 cm^−1^ was detected (Figure 2c and Figure 3a). This shift may reflect changes in the vibrational spectrum of CF bonds in TRM-PTFE macromolecules associated with changes in hexagonal packing resulting from defect formation and latent track overlap [40].

Irradiation with a stream of xenon ions led to a shift of the maximum band 1147 cm^−1^ up to 1150 cm^−1^ when the fluence of 3 × 10^10^ cm^−2^ is reached. However, at a fluence of 10^11^ cm^−2^ the maximum of the vibrational mode in question is shifted by 10 cm^−1^ in the opposite direction—up to 1140 cm^−1^ (Figure 3a, dependence 2). The shift of vibrational mode maximum of 1147 cm^−1^ toward higher wavenumbers with increasing fluence may be associated with the processes of PTFE macromolecules transitioning from hexagonal conformation to triclinic conformation [35]. On the contrary, the shift of the maximum of vibrational mode 1147 cm^−1^ towards lower values of wave numbers, observed at a fluence of 10^11^ cm^−2^, may be associated with the appearance and development of the fields of thermoelastic stresses in the ion irradiated polymer volume [41].

For the oscillation band 1206 cm^−1^ in the fluence region 10^8^–10^10^ cm^−2^, a slight shift of the maximum of the vibrational mode to the region of smaller values of wave numbers was detected, up to 1200 cm^−1^ (Figure 3a, dependence 3). When increasing the fluence to 3 × 10^10^–10^11^ cm^−2^ there was a tendency to stabilise the position of the maximum of the considered oscillatory mode in the range of 1200–1203 cm^−1^. The nature of the observed shifts of the maximum of 1206 cm^−1^ can also be explained by the development of thermoelastic stress fields and the change in the packing of macromolecules from the hexagonal conformation to the triclinic one [35,41].

The maximum of the oscillatory mode of 1238 cm^−1^ with a fluence of 10^8^ cm^−2^ underwent a huge shift in the region of lower values of wave numbers—up to 1203 cm^−1^ (Figure 3a, dependence 4). With an increase in the fluence of xenon ions to 10^9^–10^11^ cm^−2^, the considered maximum of the band 1238 cm^−1^ began to shift in the opposite direction, and when the fluence of 10^11^ cm^−2^ was reached, this maximum was observed at 1241 cm^−1^.

The change in the concentration of the centres causing the origin of the vibrational modes 1116, 1147, 1206, 1238 cm^−1^ is shown in Figure 3b. Attention is drawn to the simultaneous change in the peak areas 1147, 1206, 1238 cm^−1^ of the initial stage of irradiation, with a fluence of 10^8^ cm^−2^. The appearance of a new band 1116 cm^−1^, a decrease and an increase in the concentration of centres responsible for the nature of the vibrational modes 1147, 1206, 1238 cm^−1^ (Table 2), observed in the fluence range 10^8–^10^11^ cm^−2^, indicates the course of mutual reactions involving these centres. The products of these reactions can be centred with oscillations that have the appearance of a band at 1116 cm^−1^.

As is known [39], vibrations in the range from 700 to 900 cm^−1^ characterise the amorphous phase of PTFE. In an unaccelerated TRM-PTFE sample the registered spectral profile consists of a number of overlapping vibrational modes with maxima at ~721, ~742, ~770, ~784, ~800 and ~853 cm^−1^ whose contribution was established by approximating the observed spectral shape by a set of random Voigt functions (Figure 4a, Table 2).

The effect of accelerated xenon ions led to a decrease in the intensity of vibrational modes with maxima at 721 and 742 cm^−1^, the nature of which is associated with fluctuations in CF bonds in atomic groups –CF=O and terminal groups –CF_3_, respectively (Figure 5, Table 2). Similarly, the intensity of the vibrational modes ~770 and ~784 cm^−1^ changed due to libration fluctuations of the CF bond (Table 2). Probably, -CFO groups can occur on the surface of TPM-PTFE films both during processing on hot rolls (as a result of thermal oxidation) and during ion irradiation.

At the same time, there were no noticeable changes in the position of the maximum and intensity of wide structureless bands with maxima at ~800 and ~853 cm^−1^, which are caused by fluctuations of the –CF bond in the amorphous disordered phase and bending vibrations of the –CF_2_ groups (Figure 4, Table 2).

**Table 2 membranes-13-00101-t002:** Positions of maxima (ν_max_, cm^−1^) of vibrational modes registered in spectral ranges I–V of infra-red spectra of attenuated total reflection of films of thermoradiationally modified polytetrafluorethylene, irradiated with xenon ions with energy ~1 MeV/nucleon.

ν_max_, cm^−1^	Identification of the Oscillatory Mode	Note
721.07 ± 0.10	oscillation of the atomic group –CF=O	[33,34]
742.14 ± 0.10	oscillations of the terminal –CF_3_ groups	[33,34]
770.33 ± 0.75	libration oscillations of CF_2_ groups in the amorphous phase	[39,42]
784.77 ± 0.92	libration oscillations of CF_2_ groups in the amorphous phase	[39,42]
800 ± 0.10	oscillation of the –CF bond in the amorphous disordered phase	[36]
853.55 ± 38.02	flexural oscillation –CF_2_ groups	[43]
1116.17 ± 0.41	not identified	-
1147 ± 0.10	–CF_2_-groups symmetric stretching oscillation	[37,39]
1206.61 ± 0.67	valence oscillation of CF_2_–groups in the crystal phase	[39,43]
1237.93 ± 0.88	valence oscillation of the C-C bond	[39]

### 3.3. Surface Morphology of the Initial and Accelerated Xenon Ion Irradiated Films of ThermoRadiationally Modified Polytetrafluoroethylene

It was found by AFM that the morphology of the surface of unirradiated xenon-ion-exfoliated TPM-PTFE film is defined by lamellar structures having the form of crystallite layers, which are located perpendicular to the direction of orientation stretching of the film sample (Figure 6a and Figure 7). The crystallites had a conical shape with height y = 19–20 nm and width at the base ~16,8 nm, the distance between the tops of individual protrusions is 16.7 nm. Probably, the same distance between crystallite layers is a consequence of self-organization processes developing during the orientation of crystallites in the field of tensile stresses.

The effect of the flow of accelerated xenon ions to a fluence of 10^8^ cm^−2^ led to the complete disappearance of layered lamellar structures (Figure 6b). At the same time, the degree of heterogeneity of the surface of the ion-irradiated PTFE sample decreased from 45.3 to 22.3 nm. The observed effect is due to the process of amorphization and recrystallization of the polymer occurring as a result of ion treatment [44]. The given sizes of crystallites were in good agreement with the known estimates of the size of crystallites in sintered PTFE (~20 nm [45]), as well as with the size of crystallites calculated using the Scherer equation ~9.11 nm in TRM-PTFE samples produced by irradiation of polymer melt up to 200 kGy [23].

The effect of xenon ions with an energy of 1 MeV/nucleon at a fluence value of ~10^8^ cm^−2^ led to huge changes in the surface roughness parameters of TRM-PTFE (Table 3, Figure 7). The average roughness of R_a_ decreased by ~5.94 times, and the rms roughness of R_q_ decreased ~6.56 times compared to non-irradiated TRM-PTFE. At the same time, the maximum roughness height R_t_, and R_z_, decreased by ~7.62 and 7.72 times, respectively.

Attention is drawn to the change in the value of the asymmetry coefficient, which characterises the degree of deviation of the distribution of roughness parameters relative to the normal distribution, in the ion-irradiated TRM-PTFE with respect to the initial one. A decrease in the value of the asymmetry coefficient indicates that the characteristics of the distribution of roughness parameters in ion-irradiated up to 10^8^ cm^−2^ TRM-PTFE were close to normal.

Thus, the effect of the flow of accelerated xenon ions with an energy of ~1 MeV/nucleon to a fluence of 10^8^ cm^−2^ led to smoothing of the surface of TRM-PTFE due to the destruction of surface crystals. The proximity of the distribution of the parameters of the degree of roughness to the normal condition, found in ion-irradiated TRM-PTFE (Table 2), may be the result of the formation of surface micro-shock waves.

By analysing the surface profile of the TRM-PTFE film treated with accelerated xenon ions in several areas of the AFM image (Figure 6b and Figure 8), the average size of the track core (kern) r_core_ ≈ 2.25 ± 0.61 nm was determined.

## 4. Discussion

The detected decrease in the intensity of the 721 cm^−1^ band, which occurred with an increase in the fluence of xenon ions, may be due to the removal of surface –CFO groups as a result of radiation-chemical processes in TPM-PTFE (Figure 5). In the fluence region of 10^8^–10^10^ cm^−2^, a decrease in the intensity of the vibrational mode of 742 cm^−1^ was simultaneously observed, the origin of which was associated with the oscillations of the terminal –CF_3_ groups. With fluences 3 × 10^10^–10^11^ cm^−2^ the oscillatory mode 742 cm^−1^ was stabilized.

Probably, the decrease in the intensity of the 742 cm^−1^ vibrational band at the initial stages of TRM-PTFE radiolysis reflects the processes of crosslinking of polymer chain fragments in the LT region [34,46]. Overlapping LT can reduce the intensity of the stitching process due to balancing by the destruction process.

Consequently, when the critical fluence N_c_~3 × 10^10^ cm^−2^ was reached, the spatial conditions of the crosslinking formation process changed. It is not difficult to estimate the diameter of the region around the core of the track, in which high-temperature radiation modification occurs, which corresponds to a change in the spatial conditions for the formation of cross-C-C bonds from the mode of “isolated LT” to “overlapping LT”:(3)$$\langle d\rangle =\frac{1}{\sqrt{N_{c}}}$$

It was shown that <*d*> ≈ 57.7 nm, which is ~25.64 r_core_. Then the path length of the δ-electrons with energy above the threshold value, which ensures the predominant formation of a transverse C-C bond (hereinafter referred to as the area of localization of radiation damage), is ~ ½ <*d*> ≈ 29.0 nm.

An additional argument in favor of the formation of transverse C-C bonds is the results of the analysis of the radiation behavior of the vibrational mode 1206 cm^−1^, which characterizes the valence vibrations of carbon bonds in various atomic groups of TPM-PTFE (Table 2, Figure 3b). It was established that irradiation of the polymer with xenon ions to a fluence of 10^8^ cm^−2^ is accompanied by a significant increase in the area of the corresponding peak obtained as a result of decomposition of the contour of the spectrum of the IR-ATR in the region of 900–1400 cm^−1^ (Figure 3b). In accordance with the above, this may indicate a significant increase in the concentration of C-C bonds localized in the area of LT, which occurs as a result of the formation of crosslinking processes between fragments of polymer chains and the formation of side groups [44]. A decrease in the peak area of 1206 cm^−1^ in the fluence range of 10^9^–10^11^ cm^−2^ may be associated with the course of various radiation-chemical processes (destruction, disproportionation, formation of radicals and polyene structures, etc.) competing with the crosslinking process [34,46].

The processes leading to the reduction of the 1206 cm^−1^ band area can occur not only as a result of the exposure to the flux of accelerated xenon ions. It is known that the formation of ions, radicals and double bonds (hereinafter referred to as active centres (AC)) is also observed during plasma treatment of polymeric materials [47,48]. Due to the positive effect of these ACs on surface wetting characteristics, plasma treatment is used, among others, for hydrophilization of the surface of polymeric materials [26,48]. In contrast to plasma treatment, where only the material surface is modified, accelerated ion treatment causes formation of ACs mainly on the boundary of a nanoscale channel (LT core) with a diameter ~2.5 nm and a depth of 17.17 µm (in accordance to SRIM simulation results) extending from the surface to the TRM-PTFE volume.

Then, the LT core can be considered as an analogue of a capillary in which the processes of moisture sorption from the environment are possible. The IR-ATR method used in this work does not allow registering water absorbed in the tracks with sufficient accuracy due to the shallow depth of penetration of the probing beam. For this reason, in the next publication a detailed consideration of water sorption in LT as well as the influence of the surface concentration of LT on the wetting of TRM-PTFE films will be given.

For the theoretical calculation of track sizes, we used the $\frac{dE}{dx}$ linear energy transmission values obtained based on modelling in the SRIM program and the empirical dependence of the track core size on $\frac{dE}{dx}$ [49]:(4)$$r_{core}^{{}_{\prime }}=-4.65971+0.82633\cdot \left(\frac{dE}{dx}\right)-0.02136\cdot {\left(\frac{dE}{dx}\right)}^{2}$$
where $\frac{dE}{dx}$—total energy losses of xenon ions in interaction with carbon and fluorine nuclei, and the electronic systems represented in $\frac{KeV}{nm}$.

To calculate $\frac{dE}{dx}$ values using SRIM software, we used an estimation of TRM-PTFE density obtained on the basis of the assumption that during radiation modification elementary crystal cell of PTFE transitions from triclinic to hexagonal syngony, the process proceeds linearly with dose and the crystal part of TRM-PTFE obtained at 200 kGy dose consists of 100% hexagonal crystal cells [23]:(5)$$\rho_{50}=0.25\cdot \rho_{TRM-PTFE,\;200}+0.75\cdot \rho_{PTFE}.$$
where $\rho_{TRM-PTFE,\;200}$ and $\rho_{PTFE}$—the densities of TRM-PTFE thermoradiationally modified with a dose of 200 kGy and the initial PTFE obtained experimentally [23]. As a result of calculations using Equation (5), the density of PTFE thermoradiationally modified up to a dose of 50 kGy (≈2.20 g/cm^3^) was determined.

Using the obtained estimate of $\rho_{TRM-PTFE,\;50}$ when simulating the LT of xenon ions in a polymer thermoradiationally modified to 50 kGy, the $\frac{dE}{dx}$ value of He ions with energy ~1 MeV/nucleon $\frac{dE}{dx}$ = ~14.54 $\frac{KeV}{nm}$ was determined. Substituting this estimate into (4) it was found that $r_{core}^{{}_{\prime }}$ ≈ 2.84 nm, which is in satisfactory agreement with the size of LT observed on TPM-PTFE film surface images obtained by AFM (2.25 ± 0.61 nm).

## 5. Conclusions

This study shows that TRM-PTFE films have about three times higher hydrogen barrier properties compared to conventional PTFE. It is concluded that the nature of the effect may be related to the increase of crystallinity degree and disappearance of micropores.

It is noted that the exposure to accelerated ions does not cause destruction of the material. On the contrary, the comparative analysis of IR-ATR spectra of TRM-PTFE films exposed to fluxes of accelerated xenon ions with energy ~1 MeV/nucleon in fluence range 10^8^–10^11^ cm^−2^ demonstrated the formation of transverse carbon-carbon bonds and cross-linking of end fragments of polymer chains. It was found that C-C bond formation proceeds most intensively in the fluence range 10^8^–10^10^ cm^−2^ corresponding to the course of TRM-PTFE radiolysis in the regime of “isolated tracks”. At fluences of 3 × 10^10^–10^11^ cm^−2^ a tendency towards the reduction of cross-linking intensity due to the overlapping of latent tracks was observed.

As a result of AFM analysis of the surface of non-irradiated and irradiated TPM-PTFE films, the presence of distinct crystallites oriented perpendicular to the direction of the film’s orientation stretching was demonstrated on the surface of the unirradiated polymer. It was found that the above supermolecular structure with crystallite fragments protruding on the surface was destroyed by ion treatment of the films up to a fluence of 10^8^ cm^−2^, resulting in an approximate seven-fold decrease of the TRM-PTFE surface roughness. A possible reason for the detected effect could be the formation and propagation of microshock waves on the surface of TRM-PTFE during the interaction of xenon ions with the polymer.

As a result of mathematical processing of the results of IR-ATR spectroscopy, an estimate of the size of the region surrounding the LT xenon core in TRM-PTFE, in which intensive ion-induced processes of macromolecule crosslinking and degradation (≈ 29 nm) can take place, was obtained.

The average size of the track nucleus (2.25 ± 0.61 nm) determined by analysis of topographic images of the surface irradiated by TRM-PTFE ion streams agreed satisfactorily with the results of calculations based on simulation data in SRIM software using the existing empirical relationship described for Mg_2_SiO_4_ ($r_{core}^{{}_{\prime }}$~2.84 nm) [29,49].

Thus, TRM-PTFE films are likely to have high barrier properties and are capable of withstanding the effects of dense ionizing radiation without significant degradation. Considering that TRM-PTFE, in comparison with PTFE, has improved tribological and physical-mechanical properties, it can be concluded that this material is promising for the development of track and proton conducting membranes [26,50,51]. The effect of a significant reduction in the surface roughness of the TRM-PTFE film, irradiated by a flux of xenon ions to a fluence of ~10^8^ cm^−2^, established in this study, can be useful for the application of proton-conducting membranes based on this material in the development of graphene-polymer transistors [52].

## Figures and Tables

**Figure 1 membranes-13-00101-f001:**
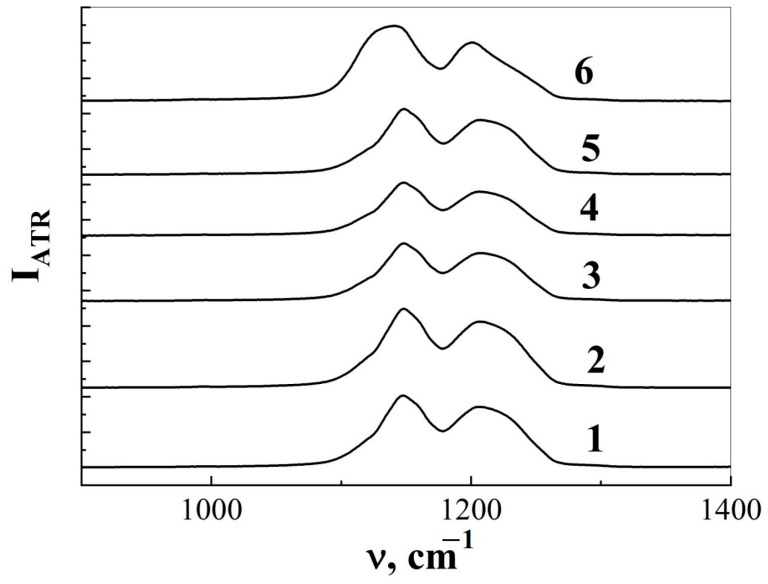
Change in the spectral shape of vibrational modes corresponding to antisymmetric and symmetric valence vibrations of the C-F bond in the infrared spectra of the disturbed total internal reflection of thermoradiationally modified polytetrafluoroethylene irradiated with xenon ion fluxes with an energy of ~1 MeV/nucleon to fluences 0 (1), 10^8^ (2), 10^9^ (3), 10^10^ (4), 3 × 10^10^ (5) and 10^11^ cm^−1^ (6).

**Figure 2 membranes-13-00101-f002:**
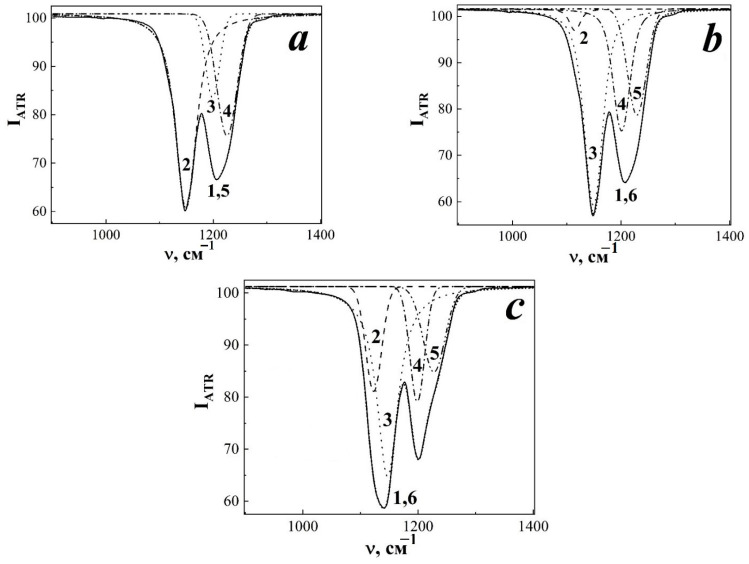
Approximation of the infrared spectra of the disturbed total internal reflection of the initial (**a**) and irradiated with accelerated xenon ions up to 10^8^ and 10^11^ cm^−2^ (**b**,**c**, respectively) thermoradiationally modified polytetrafluoroethylene. 1 (**a**–**c**) Experimental data; 2–4 (**a**–**c**), 5 (**b**,**c**) maxima of approximating random functions; 5 (**a**), 6 (**b**,**c**) result of approximation.

**Figure 3 membranes-13-00101-f003:**
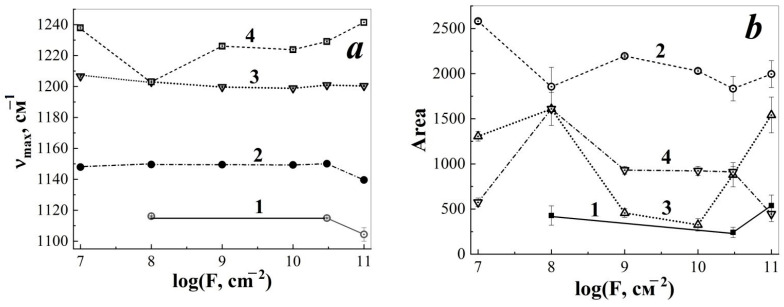
Change in the position of maxima (**a**) and peak areas (**b**) of oscillatory modes 1116 (1), 1147 (2), 1206 (3), 1238 (4) cm^−1^ in the infrared spectra of the disturbed total internal reflection of thermoradiationally modified polytetrafluoroethylene depending on the fluence of xenon ions with an energy of ~1 MeV/nucleon.

**Figure 4 membranes-13-00101-f004:**
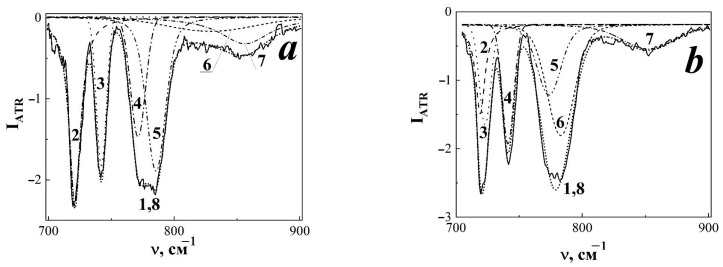
Fragment of the infrared spectrum of the disturbed total internal reflection of the initial (**a**) and irradiated with accelerated xenon ions to fluxes of 10^8^ and 10^11^ cm^−2^ thermoradiationally modified polytetrafluoroethylene (**b**,**c**, respectively) in the spectral range of 700–900 cm^−1^. 1 Experimental data; 2–6 (**a**–**c**), 7 (**b**,**c**), maxima of approximating Voigt random functions; 7 (**b**,**c**), 8 (**a**,**b**) approximation result.

**Figure 5 membranes-13-00101-f005:**
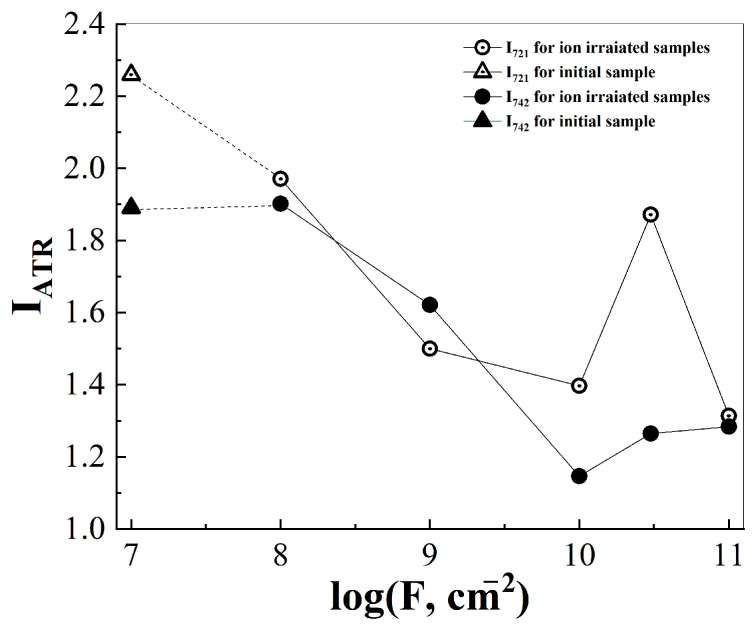
Change in the intensity of vibrational modes 721, 742 cm^−1^ in the infrared spectrum of the disturbed total internal reflection of thermoradiationally modified polytetrafluoroethylene depending on the fluence of xenon ions with an energy of ~1 MeV/nucleon.

**Figure 6 membranes-13-00101-f006:**
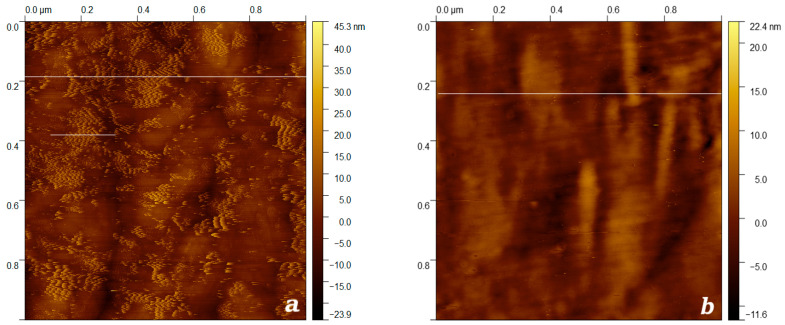
Topographic image of the surface of thermoradiationally modified polytetrafluoroethylene films before (**a**) and after (**b**) treatment with accelerated xenon ions to a fluence of ~10^8^ cm^−2^, established by atomic force microscopy (white lines highlight the measurement areas of the surface profile).

**Figure 7 membranes-13-00101-f007:**
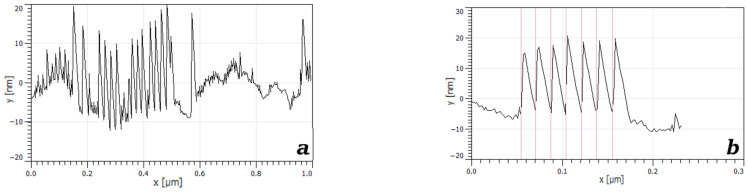
Comparison of the surface profiles of the initial (**a**) and irradiated xenon ions with an energy of 1 MeV/nucleon to a fluence of 10^8^ cm^−2^ thermoradiationally modified polytetrafluoroethylene (**c**). A fragment of the lamellar structure corresponding to a straight-line segment of white color on the topographic image of the surface of an unirradiated polymer (**b**).

**Figure 8 membranes-13-00101-f008:**
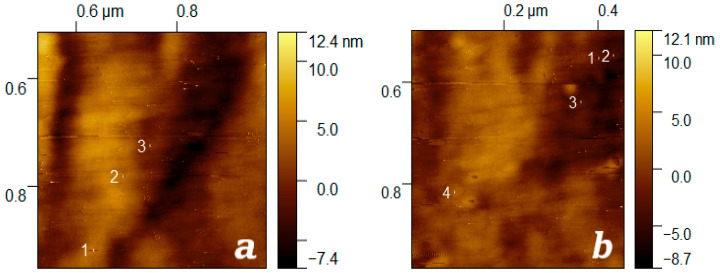
Enlarged fragments of a topographic image of the surface of a thermoradiationally modified polytetrafluoroethylene film (Figure 6b) irradiated with a stream of xenon ions up to 10^8^ cm^−2^: 1–3 (**a**), 1–4 (**b**)—tracks of xenon ions are indicated by white line segments.

**Table 1 membranes-13-00101-t001:** Comparison of the hydrogen permeability of the films of the initial and thermoradiationally modified polytetrafluoroethylene.

Title 1	PTFE			TRM-PTFE		
Results of single measurements	2.50	2.80	2.53	0.90	0.88	0.77
Hydrogen gas permeability, 10^9^ (mol·H2) · (m · s · MPa)^−1^	2.61 ± 0.17			0.85 ± 0.07		

**Table 3 membranes-13-00101-t003:** Comparison of the surface roughness parameters of the initial and irradiated xenon ions with an energy of 1 MeV/nucleon to a fluence of 10^8^ cm^−2^ thermoradiationally modified polytetrafluoroethylene.

Roughness Parameters	Basic	Fluence 10^8^ cm^−2^
Average roughness (R_a_), nm	3.21	0.54
RMS roughness (R_q_), nm	4.79	0.73
Maximum roughness height (R_t_), nm	34.30	4.50
Maximum depth of the roughness depression (R_v_), nm	15.42	2.27
Maximum height of the roughness peak (R_p_), nm	18.88	2.24
Average maximum profile height (R_z_), nm	28.13	3.64
Average maximum roughness height (R_z_ ISO), nm	22.75	2.92
Asymmetry Coefficient (S_k_)	0.74	−0.080
Kurtosis Coefficient (R_ku_)	5.04	4.096

## Data Availability

The data presented in this study are available on request from the corresponding author.

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
