# Peer review of "Thermoradiationally Modified Polytetrafluoroethylene as a Basis for Membrane Fabrication: Resistance to Hydrogen Penetration, the Effect of Ion Treatment on the Chemical Structure and Surface Morphology, Evaluation of the Track Radius"

_membranes, 2023, doi:10.3390/membranes13010101_

Round 1

Reviewer 1 Report

This work discusses the properties of thermo-radiation-modified polytetrafluoroethylene (TRM-PTFE) membranes and reported resistance to hydrogen penetration, the effect of ion treatment on the chemical structure and surface morphology, and the evaluation of the track radius.

1.      In the introduction section, proper research gaps need to be identified. Accordingly, the objectives of the work shall be stated. What is the novelty of the work??

2.      Keywords: thermo-radiation modified polytetrafluoroethylene; polytetrafluoroethylene; accelerated ions; hydrogen permeability; atomic force microscopy; IR-ATR spectroscopy; SRIM. Key wards should be free from the paper's title and single ward.  

3.      Reference no. 28 is not cited in the manuscript. Authors need to use more no references in the introduction section.

4.      Photo images of the separation unit should be provided.

5.      The conclusion section is not reporting any findings. Published paperwork is reported with a citation. This indicates that there is no novelty in work. Therefore major revision is required.

Author Response

We corrected the errors you indicated and partially rewrote the conclusions of the article. We hope that now our work will create a more complete impression for you. We did not have time to add a high-quality photo of the installation within the specified time frame, but we described in more detail all the components of installation.

Reviewer 2 Report

While thermal radiation of PTFE has  been well-studied, the author’s provide a new investigation specifically on PTFE membranes with topological manipulation in addition to new insight on surface chemistries after post-modification. The work is well-organized, appropriately cited, and offers detailed analysis necessary to support most conclusions. The work should be accepted for publication after consideration to the following points:

1. The authors should provide insight to the surface energy via wettablility of the modified membranes since this aspect should be affected by surface roughness and/or the influence of oxygen containing species as a result of PTFE degradation upon irradiation.

2. The authors should comment on the depth of penetrated radiation into the surface of the samples. 

Author Response

We corrected the errors you indicated and partially rewrote the conclusions of the article. We hope that now our work will create a more complete impression for you. In addition, we have added information that the nature of the damage as a result of ion treatment suggests hydrophilization of the material in the region of the track core. More detailed studies on taking into account sorbed water and the effect of ion treatment on surface wetting will be presented in the next work.

Round 2

Reviewer 1 Report

1.       The quality of the work is very important than the time. Therefore Authors need to provide a photo image of the Electrochem sealed cell with 25 cm3 separated gas spaces for hydrogen separation.

2.       In conclusion, they retain citations. This indicates that there is no novelty in work. Therefore, it is not possible to accept in the current state.